# Citrus Pulp Replacing Corn in the Supplement Decreased Fibre Digestibility with No Impacts on Performance of Cattle Grazing Marandu Palisade Grass in the Wet-Dry Transition Period

**DOI:** 10.3390/ani12070822

**Published:** 2022-03-24

**Authors:** André Alves de Oliveira, Eliéder Prates Romanzini, Diogo Fleury Azevedo Costa, Rondineli Pavezzi Barbero, Mariana Vieira Azenha, Josiane Fonseca Lage, Ana Cláudia Ruggieri, Ricardo Andrade Reis

**Affiliations:** 1Faculty of Agrarian and Veterinary Sciences, São Paulo State University, Jaboticabal 14884900, SP, Brazil; andre.oliveira@trouwnutrition.com (A.A.d.O.); elieder.romanzini@gmail.com (E.P.R.); ana.ruggieri@unesp.br (A.C.R.); 2Trouw Nutrition Brasil, Nutreco Group, Campinas 13080650, SP, Brazil; 3Institute for Future Farming Systems, CQUniversity, Rockhampton, QLD 4701, Australia; d.costa@cqu.edu.au; 4Institute of Animal Husbandry, Federal Rural University of Rio de Janeiro, Seropédica 23897000, RJ, Brazil; barbero.rp@gmail.com; 5Premix, Ribeirão Preto 14020670, SP, Brazil; maryazenha@hotmail.com; 6Cargill Animal Nutrition, Campinas 13091611, SP, Brazil; josiane_lage@cargill.com

**Keywords:** by-product, energy, protein, stocking rate, warm season grasses

## Abstract

**Simple Summary:**

Deterioration of forage quality as the dry season approaches has detrimental effects on the performance of cattle grazing tropical grasses unless supplementation strategies are implemented. In this study, corn was replaced by citrus pulp as an energy source to evaluate the effects of supplementation on liveweight performance and metabolic parameters of *Bos indicus* cattle grazing *Urochloa brizantha* cv. Marandu in the wet-dry transition period.

**Abstract:**

The wet-dry transition period brings unique challenges to cattle producers in the tropics as the forage quality starts to decrease and animal performance is negatively affected unless supplementation strategies are applied. Two experiments were conducted concomitantly to evaluate the performance and metabolic parameters of cattle supplemented with two different energy sources under a C4 tropical grass continuous grazing system in the wet-dry transition period. In experiment (exp) 1, the liveweight (LW) gain of 42 growing *Bos indicus* Nellore bulls allocated to 12 paddocks of *Uruchloa brizantha* cv. Marandu, in a completely randomized block design, was evaluated to compare corn and citrus pulp-based supplements offered at 0.5% LW. Metabolism was evaluated in exp 2 with eight rumen-cannulated Nellore steers in an incomplete replicated Latin square design (RLSD) 4 × 2 (steers × treatment) fed the same two treatments as in exp 1. No differences in animal performance were observed between corn or citrus pulp-based supplements. Rumen pH, ammonia nitrogen, and blood urea nitrogen were lower in the rumen of animals supplemented with citrus pulp. Despite this, there were no differences in total dry matter intake amongst the two energy sources. The neutral detergent fibre digestibility of the diet containing corn decreased, but it neither affected performance nor liveweight gain per area. The findings suggest that corn can be replaced by citrus pulp with neither detrimental effects on animal performance nor indirect effects on pastures productivity.

## 1. Introduction

Beef cattle production in the tropics and subtropics presents unique challenges to producers across the globe. Much attention has been given to the wet [1] and drier months of the year [2] simply because there is a clear distinction in quantity as well as the quality of forage produced in these contrasting seasons. However, the transition periods may be equally important with specific limitations on forage variables that can compromise animal performance and productivity. The transition is also an important period to adjust management practices and start adapting animals to supplemental feedstuffs and to a feedlot finishing phase [3]. These supplements may result in substitution effects due to changes in ingestive behaviour and have direct effects on the quality and quantity of the final diet (i.e., supplement + grass) [4]. Nevertheless, the right adoption of supplementation strategies promotes improved animal performance and productivity in tropical grazing systems despite the reduced growth of forage in periods of less appropriate climate conditions [5]. Moreover, the utilization of human-inedible ingredients in supplementation strategies may help promote sustainable intensification of beef production [6]. The ruminant’s ability to upcycle these by-products into high-quality human-edible protein has been highlighted [7]. In this context, the utilization of by-products such as citrus pulp replacing corn, an ingredient heavily utilised in human nutrition, represents an important factor in increasing sustainability on the intensification of ruminant production systems. Costa et al. [1] found that a citrus pulp-based supplement helped animals maintain performance under high grazing pressure in a rotational system of *Urochloa brizantha* during the wet season, whilst in the current experiments, corn and citrus pulp-based supplements were used in the transition period, when both quality and amount of forage produced decrease.

We hypothesized that animal performance and metabolism of growing *Bos indicus* Nellore cattle are positively influenced by additional energy provided by supplementation during the wet-dry transition period and that no major differences would be observed between corn or citrus pulp-based supplements.

## 2. Materials and Methods

### 2.1. Experimental Area and Previous Management

The two experiments were conducted in an area of 13.8 hectares of Marandu palisade grass (*Uruchloa brizantha* cv. Marandu) at the Forage Crops and Grasslands section of Sao Paulo State University (Unesp), “Júlio de Mesquita Filho” campus in Jaboticabal, São Paulo, Brazil, 21°15′22″ S, 48°18′58″ W, 595 metres above sea level. The area was divided into 12 paddocks (six of 1.0 ha and six of 1.3 ha) previously managed under continuous grazing with variable stocking rates during the wet season from 7 January to 30 April 2011. Fertilization was applied in the wet season with a total of 188, 28, and 16 kg of nitrogen, phosphorus, and potassium per hectare, respectively, resulting in pastures averaging 8.2 t of herbage mass dry matter (DM), 15.45% crude protein (CP) and 60.73% of total digestible nutrients (TDN). No further fertilization was used during the wet-dry transition period between May and August 2011, when the experiments described in the present study were carried out.

### 2.2. Experimental Treatments, Feeding, Grazing, and Liveweight Measurement Protocols

The treatments in both experiments 1 and 2 consisted of corn or citrus pulp-based supplements (Table 1) offered at 0.5% liveweight (LW) to cattle subjected to continuous grazing on Marandu palisade grass pastures. The grazing management followed a continuous grazing protocol used in the previous wet season; however, the stocking rate was fixed throughout the wet-dry transition period in the current experiments.

In exp 1, a total of 42 growing Nellore bulls (307 ± 7.97 kg LW) of approximately 17 months of age were allocated in a completely randomized block design to 12 paddocks (four animals into six paddocks of 1.3 ha and three animals in six 1.0 ha paddocks) to evaluate animal performance, TDN intake, and digestibility. Initial LW was used for animal distribution into paddocks to balance LW difference amongst blocks (two paddocks and two supplemental treatments per block). All paddocks had access to an area with water and a bunk where the animals were group fed the supplements daily at 10:00 h. The experiment lasted 85 days, from 15 May 2011 until 7 August 2011, and it was divided into three periods matching LW measurements.

In exp 2, evaluation of metabolic parameters was conducted using eight rumen-cannulated Nellore steers (303 ± 20 kg LW) allocated to eight of the 12 paddocks (simultaneously to the animal performance trial) in an incomplete 4 × 2 replicated Latin Square design consisting of four replicates (steers) and two treatments (energy source). The liveweight of cannulated steers was taken into consideration to calculate stocking rates and supplement allocation. Each experimental run lasted 14 days, with the first five days used for adaptation and the last nine for sample collections. During the adaptation, animals were offered daily one of the two supplements at 0.5% LW (as fed basis) according to treatment. However, during the collections, supplements were put directly into the rumen cannula and animals were prevented from accessing the bunk area to guarantee adequate consumption of the allocated amount per animal.

### 2.3. Experimental Periods, Forage Collections, Morphological and Chemical Evaluations

The study in exp 1 was divided into three experimental periods 15 May 2011 and 10 June 2011 (i.e., Period 1), 11 June 2011 and 10 July 2011 (i.e., Period 2), and 11 July 2011 up to 07 August 2011 (i.e., Period 3). The latter represented the days between LW measurements. The collections of forage were conducted at the start of each of the three experimental periods, coupled with LW measurements. Prior to the determination of forage mass, sward height was measured in 80 places across each paddock. Forage was collected from four distinct sites where the sward height was similar to the average sward height value found for the paddock being sampled. Forage was harvested at ground-level using a metal frame of 0.25 m^2^ as in Barbero et al. [9]. Forage samples were collected into a plastic bag, identified, weighed, and immediately transferred to the lab for evaluation of morphological characteristics. Briefly, a subsample of approximately 500 g was separated into green leaves, green stem, and dead material and placed in a forced-air oven (55 ± 5 °C, for 72 h) for estimates of DM contribution in the sward for each plant morphological component. A separate 500 g subsample was dried as above, ground in a mill model 4 (Thomas-Wiley Laboratory, Minneapolis, MN, USA) following method #934.01 [10] and used to estimate DM forage mass per hectare.

In addition, for the determination of forage chemical composition, 20 hand-plucked samples were collected from each experimental paddock and processed as indicated by Barbero et al. [9]. The determination of DM, organic matter (OM), and ether extract (EE) were performed according to AOAC [10]. Crude protein was determined via combustion using a Leco F528 N analyser (LECO Corporation, St. Joseph, MI, USA) and multiplying N by 6.25 for CP concentration. Neutral detergent fibre (NDF) and acid detergent fibre (ADF) concentrations were determined using the method described by Delevatti et al. [11] using an ANKOM 2000 Fibre Analyser (ANKOM Technologies, Macedon, NY, USA) without sodium sulphite or α-amylase. The carbohydrate fractions were determined according to Sniffen et al. [12]. Lignin concentration was measured after hydrolysing the cellulose in ADF residues with 72% sulfuric acid (H_2_SO_4_). Forage residues of NDF and ADF were recovered and analysed for their CP and ash contents in order to determine the NDF corrected for ash and protein (NDFap) according to Delevatti et al. [11]. In addition, N fractions were determined by the method described by Licitra et al. [13].

In addition to the aforementioned analyses, the quantification of non-fibrous carbohydrate (NFC) in the supplements and the NFC in the forage samples were calculated by subtracting total DM (100%) by ash, NDFap, CP and EE levels. For digestibility analysis, the indigestible NDF (iNDF) concentrations were determined using the method described in Barbero et al. [9], with in situ incubation for 240 h. Total digestible nutrient contents of forage and supplement samples were estimated according to the National Research Council [8].

### 2.4. Metabolic Parameters

The evaluations of metabolic parameters in exp 2 were simultaneously carried out with the performance study in exp 1, with collections occurring across all experimental periods. To estimate faecal excretion, an external marker LIPE^®^ (lignin isolated, purified and enriched of *Eucalyptus grandis*; Produtos de Pesquisas Simões Saliba, Florestal, MG, Brazil) was administered daily via the rumen cannula in a capsule (500 mg) for six days, starting on day 6 of each experimental period. These capsules were administered at 11:00 h. Faeces were collected twice per day on day 9 (07:00 and 13:00 h), day 10 (09:00 and 15:00 h), and day 11 (11:00 and 17:00 h). The faeces collections occurred when the animals were observed defecating. The collections were carefully carried out in the paddocks discarding the parts that touched the ground to avoid contamination. The faeces were then oven-dried (55 ± 5 °C for 72 h), ground in a mill model 4 (Thomas-Wiley Laboratory, Minneapolis, MN, USA) and combined for each steer and period. Approximately 15 g of bulk faeces from each sample was sent to the Federal University of Minas Gerais (UFMG) to estimate faecal DM production [9]. Total DM intake was estimated using iNDF as an internal marker. The iNDF in the hand-plucked supplement and faecal samples were determined as described by Barbero et al. [9]. The use of hand-plucked samples was adopted to simulate what an animal would be consuming, as indicated by Barbero et al. [9]. To estimate apparent digestibility coefficients, faeces samples were analysed to quantify DM, ash, CP, NDF, and EE levels. With the total nutrient intake, from forage and supplements, and excretion of these nutrients into faeces, total apparent digestibility (TAD) was calculated using the following equation:TAD = (TDMI − FE)/TDMI;
where TAD = total apparent digestibility (%); TDMI = total dry matter intake (kg per day) and FE = faeces excretion (kg DM per day).

On day 12, blood samples were collected directly from the jugular vein using vacutainers and were placed in ice for approximately 20 min until processing. Samples were centrifuged for 15 min at 15,000× *g*. Plasma fractions were placed in Eppendorf tubes and frozen at −15 °C until analysis of blood urea N (BUN). Analysis was performed using commercial kits from Labtest (Lagoa Santa, MG, Brazil). The total BUN was calculated by multiplying blood urea in plasma by 0.4667, which corresponds to the percentage of N present in the urea molecule.

Rumen fluid pH and rumen ammonia N were measured using rumen fluid collected on day 13, at 0, 3, 6, and 18 h after supplementation. The time zero was considered before supplementation, which occurred at 11:00 h. The remaining collections represented the hours after supplementation. Rumen fluid was collected in the solid-liquid interface of the rumen. These samples were then filtered using triple-layer gauze, and the pH was measured using a portable digital peg MA522 model (Marconi laboratory equipment, Piracicaba, SP, Brazil). The equipment was first calibrated with buffer solution (pH 7.0 and 4.0). After the pH was measured, 40 mL of rumen fluid was separated, and 1 mL of sulphur acid (1:1) was added, and then frozen at −20 °C. These samples were used for the analysis of rumen ammonia N following the Kjeldahl method [10] with 2 N KOH as indicated by Fenner [14].

The N balance and efficiency of microbial synthesis (EMS) were calculated considering the basis for nutrient balance research in mammals as described in Maynard and Loosli [15], in which N animal products = N feed − N excreted. This method was adapted in the present study using spot urine sampling to estimate urinary excretion of nitrogenous compounds and the following equation to estimate N balance:N balance (g/day) = Intake N (g/day) − Faecal N (g/day) − Urinary N (g/day).

The EMS were measured according to Barbosa et al. [16], where total purines absorbed at (Y, mmol/day) were calculated from the excretion of purine derivatives (X, mmol/day), by the equation:Y = 70 X/0.93 × 0.137 × 1000
where 70 = contend of N purines (mg/mmol), 0.93 = true microbial purine digestibility and 0.137 = ratio N purine:N total bacteria

The N microbial was expressed in grams per day and from this result the microbial protein synthesis was calculated by multiplying it by 6.25 and then the EMS per kg of TDN consumed.

### 2.5. Liveweight Performance, Nutrient Intake, and Digestibility

Evaluation of liveweight performance as average daily gain (ADG) was achieved based on the LW at the start and at the end of each experimental period in exp 1. Experimental animals had to undergo a 14 h feed and water deprivation prior to initial and final LW measurements at 07:00 h at the beginning and end of the experiment but did not fast on the intermediate weighing on 10 June and 10 July 2011 which was used to allow stocking rate calculations for individual periods. 

Dry matter intake was evaluated in the second period of exp 1 using 36 animals, i.e., six repetitions per block, and three markers: faecal production, total dry matter intake, and supplement intake. Individual supplement intake was estimated using titanium dioxide (TiO_2_) as an external marker [9]. The TiO_2_ was mixed into the supplement, with a daily proportion of 10 g per animal. For calculations of intake and digestibility, nine days were used in the faecal evaluation, six to adapt and three for faecal collections. Faeces were processed, and TiO_2_ concentrations were measured by an atomic-absorption spectrophotometer. Supplement intake and digestibility were determined by evaluating the level of the marker in the supplement and in the faeces. Forage intake was calculated by subtracting total DM intake and supplement intake.

### 2.6. Stocking Rate, Productivity per Area and Forage Allowance

Productivity as total gain per hectare (GPH) was calculated considering the mean ADGs of animals in a particular treatment (i.e., 25 per treatment) and the number of animals in each paddock, according to the evaluation period. Productivity was measured in kg per hectare per day because the number of days in each period was not equal. The LW of cannulated animals used in exp 2 were also included in the calculations of stocking rate and GPH.

The stocking rate was calculated using the number and average ADG of animals of both exp 1 and 2 in each paddock, considering an animal equivalent (AE) equal to 450 kg LW.

Forage DM allowance and green leaves DM availability were calculated and expressed in kg of DM per kg LW as described in Barbero et al. [9]. Briefly, the total weight of the forage collected per paddock from four distinct sites multiplied by the forage DM content was used to estimate DM forage mass per hectare. Similarly, the total DM of leaf components was used to calculate green leaves DM availability.

### 2.7. Statistical Analysis

In exp 1, the experimental design used for comparisons of animal performance data and forage variables was a completely randomised block design with 2 treatments (two supplements), totalling six repetitions (12 paddocks, 6 per treatment/block). Animal variables were collected individually, but the experimental unit was represented by all individuals in a single paddock. All variables were analysed using MIXED Procedure in SAS (SAS Institute Inc., 2022, version 9.4, Cary, NC, USA). Similarly, relationships of the variables forage mass, chemical composition, forage availability and stocking rate were analysed considering repeated measurements per block and considering the collection dates in the model. For each variable, the best covariance structure was considered to have the lowest Akaike information criterion values. The models included the fixed effects of supplements, time, and the interactions amongst them. A 5% level was adopted to assess statistical significance between all comparisons.

In exp 2, an incomplete RLSD of 3 × 2, with three periods and two treatments, was used for metabolic variables. These metabolic parameters were analysed as repeated measurements using GLM in SAS (SAS Institute Inc., 2022, version 9.4, Cary, NC, USA). For rumen ammonia N and rumen fluid pH were both considered 0, 3, 6, and 18 h after supplementation; and for BUN, 0 and 6 h. The models included the fixed effects of supplements, time, and the interactions amongst them. Individual animals and the experimental periods were considered as random effects, and a Student *t*-test was used for comparisons between supplements and time effects. Statistical significance was set at a 5% level.

## 3. Results

Forage mass, stocking rate, and availability of green leaves did not differ (*p* = 0.22, *p* = 0.26 and *p* = 0.96, respectively) between treatments (Table 2). The measured herbage mass was above 7970 kg DM/ha for both treatments. The stocking rate was above 2.70 AE/ha, and the availability of green leaves was 0.88 kg DM/kg LW. There were no differences in the chemical composition of herbage samples in DM, OM, CP, NDF, ADF, Lignin and TDN (*p* > 0.14) between paddocks during the evaluation. The mean chemical composition was 39.43; 90.66; 10.83; 63.33; 33.24; 4.03 and 57.42% for DM, OM, CP, NDF, ADF, Lignin and TDN, respectively.

The DM and nutrient intakes did not differ (*p* > 0.56) between supplemental treatments (Table 3). Dry matter intake was above 6.40 kg/day for all treatments, and the CP intake was not affected by different energy sources (*p* = 0.56). The TDN intake (*p* = 0.62) was similar between the supplements evaluated, i.e., 3.89 and 3.76 kg/day for animals fed with ground corn and citrus pulp-based supplements, respectively.

A significant difference (*p* < 0.01) was found between supplement and time (Figure 1) when evaluating rumen fluid pH and rumen ammonia N. Regarding the effects of the supplements, differences were observed for all variables, with a higher rumen fluid pH value (*p* = 0.03) in animals supplemented with corn. The time effect on rumen pH value was linear in corn supplemented animals and quadratic in animals fed the citrus pulp-based supplement. The rumen ammonia N was lower (*p* < 0.0001) in animals fed citrus pulp than animals fed ground corn as an energy source.

No differences were observed for all variables around N balance (*p* > 0.50) and EMS (*p* > 0.40) between both supplemental treatments evaluated (Table 4). Blood urea N concentrations were lower (*p* < 0.01) in animals fed citrus pulp (14.75 mg/dL) than in the ones supplemented with ground corn (17.80 mg/dL).

The total DM, forage and supplement intake did not differ (*p* > 0.25) between supplemental treatments (Table 5). From all the nutrients evaluated, only the intake of non-fibrous carbohydrates differed (*p* = 0.03), being lower for ground corn (1.66 kg/day) in comparison to citrus pulp-based supplementation (1.91 kg/day). Dry matter intake values were above 2% of LW for all treatments, where forage intake was close to 1.6% LW, and supplement intake was close to 0.5% of LW. The in vivo apparent digestibility (IVD) did not differ (*p* > 0.26) between treatments for DM, OM, EE, and CP. Despite this, the IVD of NFC and NDF fractions were different between supplements evaluated (*p* = 0.02 and *p* = 0.01, respectively). The IVD of NFC animals fed citrus pulp-based supplement had lower apparent digestibility (81.42%) in comparison to animals fed ground corn supplement (87.95%), and the results for IVD of NDF showed higher apparent digestibility for animals fed citrus pulp (53.74%) in comparison to animals fed ground corn (49.28%).

Animal performance was similar (*p* > 0.50) between treatments (i.e., ground corn and citrus pulp-based supplements; Table 6). As planned, the initial LW (iLW) was similar and averaged 306.43 kg. Final LW (fLW), ADG, and GPH also did not differ between treatments, averaging 369.31 kg, 0.761 kg/day and 2.75 kg/ha per day, respectively.

## 4. Discussion

As the dry season approaches, pastures tend to decrease in quality and amount of forage being produced, which can have detrimental effects on the performance of cattle grazing on tropical grasses. Cattle in the current experiments were grazing during the transition between the wet and dry seasons and had the same allowance of forage with similar conditions of sward structure and chemical composition. Overall, nutrient intakes were the same between supplemental treatments, most likely because the experimental animals were offered a basal diet (Marandu palisade grass) of similar characteristics. The crude protein and the TDN content of the forages were not different between treatments, and the pastures had equal green leaf allowances. Despite this, the different energy sources used in the two supplemental treatments tested influenced metabolic parameters. The minimum rumen pH value was observed 6 h after citrus pulp supplementation (i.e., 6.32); however, it was still above 6.2, which has been indicated as the point in which cell wall degradation is compromised [17]. The reduction in rumen pH in animals supplemented with citrus pulp in comparison to the corn-based supplement may be related to the carbohydrate composition of these sources. Corn has a high content of starch, while the citrus pulp is rich in pectin. Both (starch and pectin) are classified as B1 fractions of carbohydrates and have fast digestion rates (10 to 50%/h; [18]). Although, pectin is quickly degraded in the rumen whilst ground corn, on the other hand, has high insoluble starch content, which degrades slower than pectin [12]. According to Owens and Goetsch [19], this reduction occurs between 30 min and 4 h after feeding. Therefore, most likely, the higher pectin digestion rate led to a decreased rumen pH.

The rumen ammonia N reduction observed three hours after citrus pulp supplementation suggests that there may be increased growth of bacteria that used pectin in their life cycle. Microorganisms that use ammonia as a source of N grow quicker than others that use cellulose or hemicellulose [20]. Our results are consistent with those reported by Ariza et al. [21], that found higher volatile fatty acids (VFA) in diets rich in citrus pulp compared to starch. When citrus pulp is used as an energy source, it reduces rumen ammonia N and increases microbial protein synthesis, although, in this study, no differences in EMS were observed. Ben-Ghedalia et al. [22] suggested that citrus pulp produced favourable conditions in the rumen for the use of other carbohydrates, such as cellulose, by rumen microorganisms compared to diets using starch as the energy source because of differences in fermentation pathways. Costa et al. [23] compared different sources of NFC using in vitro techniques and observed that starch decreased the digestibility of NDF of low-quality forages whilst pectin had no detrimental effects. In exp 2, animals fed ground corn had lower IVD of NDF than animals supplemented with citrus pulp. This corroborates with Costa et al. [24], who observed that microorganisms that use pectin use rumen ammonia N faster than microorganisms that use cellulose or hemicellulose as substrates and the addition of pectin could potentially cause a reduction in the growth rate of specific microorganisms if ammonia were limiting and therefore affect the digestibility of NDF. Despite this, the mean ammonia N concentration along the day in animals supplemented with citrus pulp was above 20 mg/dL, which indicated that there was no limitation in cellulolytic bacteria growth [25]. Lazzarini et al. [26] suggested values above the 15 mg/dL as ideal values to maximize intake of low-quality forages.

Both energy-protein supplements increased rumen ammonia N three hours after supplementation, mainly due to the existence of urea in the composition of these supplements, resulting in higher BUN concentrations six hours after supplementation. The concentration of BUN is correlated with N intake [27] and ammonia N concentration [28]. In the current experiment, N intake was not different between animals fed both supplements, and it is likely that the lower ammonia N concentration in the rumen of animals supplemented with citrus pulp led to lower BUN levels observed in exp 2. This could be related to a higher utilisation efficiency of N. Differences amongst individuals in the efficiency of N utilisation are expected [29,30], but the experimental treatments in the current trials were randomly distributed in blocks using paddocks as the experimental units to avoid these issues.

Valadares et al. [31] evaluated four levels of CP (7.0, 9.5, 12.0 and 14.5% CP) in diets of steers and reported levels of BUN as 8.1, 9.1, 15.7 and 19.5 mg/dL, respectively. The authors concluded that plasma urea levels between 13.52 and 15.15 mg/dL correspond to maximum EMS and probably would be the limit in which zebu steers experience protein losses. The BUN value was closer to 15 mg/dL in animals fed with citrus pulp supplement in the current study. However, corn-supplemented animals had BUN levels higher at approximately 19 mg/dL. The latter value could result in protein losses caused by increases in N excretions. However, the results of N balance did not indicate any difference between supplements. In contrast, Harmeyer and Mertens [27], reported that urinary N is mainly influenced by its concentration on blood plasma. Additionally, energy is spent because of ATP utilization in urea synthesis by the liver [20].

In the current study, animals supplemented at 0.5% LW gained in average 0.76 kg per day, similar to the liveweight performance of growing bulls under a rotational grazing system observed by Costa et al. [1], using the same supplement allowance and management based on sward heights of 25 cm at the start and 10 cm post-grazing. Based on the equations of Marcondes et al. [32], grazing Nellore bulls of 350 kg of LW with a DM intake of 2.2% of LW and consuming about 4.75 kg TDN/day would have an ADG of 1.0 kg. The average intake values of DM (2.12% of LW) and TDN (4.41 kg/day) were below it, and as a result, the ADG observed in exp 1 was also lower. This can be explained by the lower quality of the pastures in the transition period, which inhibits energy intake despite the high forage allowance; the high stem and dead material proportions most likely affected DM intake. In the work of Costa et al. [1], differences in forage allowance resulted in approximately 0.1 kg difference in ADG between animals in two treatments fed the same supplement at 0.5% LW. Ground corn fed at 0.0%, 0.3%, 0.6%, and 0.9% LW to growing beef cattle under a rotational grazing system of Marandu palisade grass resulted in linear increases in ADG, stocking rates and in gain per hectare, or the amount of beef produced per area [3].

The increase of fibrous compounds such as NDF, ADF and lignin could result in negative responses on intake. These compounds are directly related to the capacity of the feed in providing rumen fill that directly affects voluntary intake [33]. Despite the lower iNDF of the citrus pulp-based supplement and lower in vivo NDF digestibility in cannulated steers fed corn, there were no effects on DM intake between supplemental treatments.

The energy intake was limited, firstly due to the sward structure and non-nutritional factors, e.g., the density and physical properties of grass stems, and secondly because of nutritional factors associated with lower TDN values and a high indigestible fibre proportion in the forage in the current experiments. The CP values in the forage were above 8%, which is the minimum value to maintain an ideal microbial growth for fibre degradation [26].

Total N values in forage samples in the current work indicated that N was not limiting microbial growth. The intakes of CP observed here were above 1.13 kg CP/day that, according to Marcondes et al. [32], would allow ADGs reaching 1.0 kg of LW. This leads to the assumption that energy, as TDN intake, was the limiting factor for liveweight gain. Forage allowance and structure can limit forage intake, but the allowance was the same in all treatments. Costa et al. [1] fed a citrus pulp-based supplement to growing grazing bulls in a rotational system of Marandu palisade grass under two grazing pressures, and the extra energy through supplementary feed was found effective to overcome the deleterious effects of higher grazing pressure, and higher gains were achieved, but these were still under 1.0 kg ADG. The metabolizable energy in corn was tested against citrus pulp in diets of finishing cattle in which ADG of cattle fed citrus pulp was found equal to corn in 50% replacements and a linear decrease in values at 75 and 100% replacement [34]. In summary, supplying energy-protein supplements to grazing animals is important to either increase or help minimise the deleterious effects on animal performance, because, in addition to protein deficiency, energy may be deficient, causing a linear reduction in TDN levels in the final diet. In the current work, this can be validated by the high ratio between CP intake and digestible organic matter (DOM) from the diet. The mean value obtained was 298.7 g CP kg/DOM, which is higher than the critical value 210 g CP kg/DOM indicated by Poppi and McLennan [35] as the threshold for N losses. Detmann et al. [36] highlighted one of the main benefits of supplementation as the increase in N levels for improved animal metabolism, where positive effects can be found even when using average-to-high quality forage. However, Ramalho et al. [37] and Costa et al. [3] found that protein supplement had no incremental effects over energy supplementation on ADG of growing bulls grazing well managed tropical grasses during the wet season.

Supplementation with low and medium levels (below 0.3% of BW) during the wet season can help meet the protein requirements and keep microorganism activity for more efficient herbage degradation [11]. However, to obtain high gains, the addition of energy from supplements is required and should allow higher performance levels independently of forage availability due to limitations associated with the sward structure.

The nutrient digestibility of the final diet, i.e., basal forage and supplement, did not change between treatments, except for NFC and NDF values. The higher proportion of NFC in the citrus pulp supplement provided both higher intake and digestibility. Some authors reported that lower total digestion of NFC when corn is used is caused by the high starch content and the protein matrix involving the molecule interferes with rumen degradation [33]. However, these authors emphasised that a portion of starch can be degraded in the large intestine. This digestion in the large intestine would potentially improve microbial activity but in a site of the digestive tract of the host animal that does not allow it to use the protein of the microbial cells.

The reduction in NDF apparent digestibility with corn supplementation can be associated with competition between rumen microorganisms. A study mentioned that some authors had observed inhibition of fibrolytic enzyme activity with the presence of buffer starch [33]. This is associated with inhibitor compounds released by microorganisms that degrade starch.

Forage mass and green leaf allowance did not differ between paddocks used in this work, and neither did liveweight performance between supplemental treatments. A study by Euclides et al. [38] showed that changes in ADG of animals grazing signal grass *Urochloa decumbens* were not explained by changes in forage chemical composition alone. Rather, their work showed a significant correlation between ADG, chemical composition and the sward structure, which demonstrated to be more important than chemical composition in terms of forage intake of animals subjected to different grazing intensities.

Oliveira et al. [39] evaluated the performance of heifers during the wet-dry transition and dry season and reported that changes in performance were caused by reductions in green herbage availability. The latter authors claimed that even with a reduction in stocking capacity between the seasons [wet, 5.32 animal equivalent (AE)/ha and dry, 2.82 AE/ha), and high forage allowance during the dry season, animals supplied with protein supplements at 0.5% of LW or mineral mix had a reduced ADG as green mass availability decreased. However, animal performance was improved for supplemented animals in comparison to animals on mineral mix alone. Costa et al. [1] observed a 0.43 AE/ha increase in stocking rates in growing bulls grazing Marandu palisade grass pastures during the wet season and fed a citrus-pulp based supplement at 0.5% LW in comparison to non-supplemented animals from the same cohort.

In the current work, the ADG of experimental animals in exp 1, i.e., 0.76 kg/day, was in accordance with DM and TDN intakes observed. Although, as the availability of green leaves decreased, both supplemental treatments seemed to meet the animals’ requirements allowing satisfactory ADG during the transition period. According to the National Research Council [8], when more than 1.00 kg of supplement is supplied daily to animals, forage intake can decrease because of substitution effects. However, when used as a tool in the intensification of beef production systems, this decrease may be desired, provided it increases animal performance due to a higher digestible energy intake, or when it results in higher stocking rates without changes in animal performance. This is discussed in the review by Santos et al. [5] and is corroborated by the findings of Sales et al. [40], that evaluated growing supplement levels (0.3, 0.4 and 0.5% of LW) during the wet-dry transition period and reported a linear reduction in forage intake and a positive linear effect of energy levels on ADG (0.664 kg/day) and final LW. The authors highlighted that increases in ADG caused by the substitution effect should be evaluated because it results in higher costs, which depending on handling type and system objectives, may not be profitable.

All results showed that utilization of citrus pulp replacing corn in the supplementation of beef cattle during the wet-dry transition period represented a good strategy to increase sustainability in the intensification of a grazing production system. Mixed crop-livestock systems involve interactions between livestock and their feed based on pastures and annual or perennial food crops that result in several by-products with high potential to be used in ruminant nutrition at pastures or feedlot systems [41]. The use of human-inedible by-products and grassland areas by ruminants decreases the competition between ruminants and humans for food and increases the availability of high-quality animal protein for human consumption [6,42]. Data from our experiments confirm that the supplementation of animals grazing on low-quality forage can increase DM intake, the efficiency of nutrients utilization and ADG. The use of these supplementation strategies can reduce slaughter age and consequently methane emissions in comparison to non-supplemented animals [41,43,44]. As indicated by Cappai et al. [45], circulating parameters of clinical importance may indicate homeostasis perturbations. The latter authors highlighted the importance of body fluid distribution and how it links to nutrient and energy availability to organs and tissues. In their work, the concept was applied to pregnant does and showed differences amongst individuals delivering single or twin kids. In the current work, the supplement type was a factor potentially affecting the metabolic profile of the animals. Therefore, differences in animal performance were evaluated in parallel to key metabolic parameters of relevance to ruminant nutrition.

The season in the year, e.g., wet or dry, was shown to affect rumen function in grazing cattle more than forage type [46]. The animals in the current study were grazing Marandu palisade grass pastures in the wet-dry transition period. Changes in diet quality are expected to occur with shifts in temperature, lower rainfall events, and fewer daylight hours. In this context, supplementation strategies need to consider the overall diet quality, supplying the limiting nutrients restricting animal performance. In addition, the selected strategies need to consider the increase in costs of production, which could compromise profitability [47]. The use of by-products is preferred in this context due to not generating concerns surrounding feed-food competition [7].

## 5. Conclusions

Despite existing differences in metabolic parameters, the dry matter intake, N balance and efficiency of microbial synthesis of cattle supplemented with either ground corn or citrus pulp at the evaluated level were not affected. Most importantly, liveweight performance was similar between treatments indicating that any of these energy sources could be used during the wet-dry transition period. The use of citrus pulp, a human-inedible feed has some advantages from the sustainability viewpoint; however, producers can choose which supplementation strategy to adopt based on overall system evaluation, which may include supplement costs.

## Figures and Tables

**Figure 1 animals-12-00822-f001:**
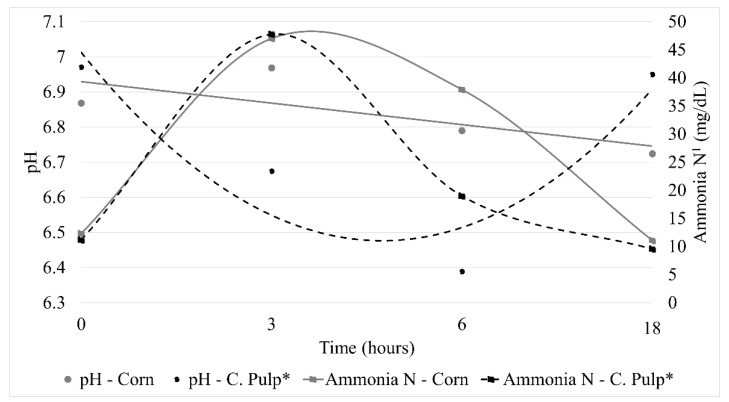
Rumen parameters at different sampling times (0, 3, 6 and 18 h) of rumen-cannulated Nellore steers continuously grazing *Uruchloa brizantha* cv Marandu pastures and fed corn or citrus pulp-based supplements during the wet-dry transition period; ^1^ Ammonia N: rumen ammonia nitrogen; C. Pulp* = citrus pulp.

**Table 1 animals-12-00822-t001:** Ingredient and chemical composition of supplements in experiments 1 and 2.

Items	Treatment	
	Corn	Citrus Pulp
Ingredient composition (% DM ^1^)		
Ground corn	46.94	-
Citrus pulp	-	53.74
Cotton seed meal	39.09	39.80
Urea	3.09	3.02
Mineral mix ^2^	10.88	3.43
Chemical composition (% DM)		
Dry matter	92.27	91.14
Organic matter	84.23	89.18
Ash	15.76	10.81
Ether extract	2.73	1.05
Crude Protein	33.95	33.59
NDF ^3^	26.65	30.94
NDFap ^4^	22.42	26.16
iNDF ^5^	10.44	9.19
ADF ^6^	22.81	26.93
Lignin	3.57	8.73
TDN ^7^	69.90	64.66

^1^ DM = dry matter; ^2^ Mineral mix compositions: Corn supplement—1.87% NaCl; 0.0085% CuSO_4_; 3.14% Caulin; 5.8% Calcite 37; 0.022% Monensin 200; 0.0019% Ca(IO_3_)_2_; 0.004% NBM 2.2; 0.0011% Moonstone; 0.0053% ZnO. Citrus pulp supplement—1.76% NaCl; 0.0085% CuSO_4_; 1.6% Calcite; 0.022% Monensin 200; 0.0021% Ca(IO_3_)_2_; 0.0006% MnO; 0.0013% Moonstone; 0.0055% ZnO; ^3^ NDF = neutral detergent fibre; ^4^ NDFap = NDF corrected to ash and protein; ^5^ iNDF = indigestible NDF; ^6^ ADF = acid detergent fibre; ^7^ TDN = Total digestible nutrients: estimated according National Research Council [8].

**Table 2 animals-12-00822-t002:** Forage mass, stocking rate, green leaf allowance, and chemical composition of pastures in the wet-dry transition period in paddocks of both supplemental treatments.

	Treatment			
Variable	Corn	Citrus Pulp	SEM	*p*-Value
Forage mass (kg of dry matter/ha)	8305	7655	476.9	0.22
Stocking rate (animal equivalent/ha)	2.83	2.71	0.14	0.26
Green leaf allowance ^1^ (kg DM/kg LW)	0.88	0.88	0.11	0.96
Chemical composition (% DM)				
Dry matter	39.91	38.96	0.13	0.82
Organic matter	90.66	90.65	0.21	0.98
Crude protein	11.04	10.62	0.26	0.67
Neutral detergent fibre	63.45	63.20	0.39	0.83
Acid detergent fibre	33.33	33.15	0.35	0.85
Lignin	4.22	3.84	0.12	0.14
Total digestible nutrients ^2^	57.68	57.16	0.20	0.45

^1^ Green leaf available in kg of dry matter per kg of liveweight; ^2^ Total digestible nutrients estimated according to National Research Council [8]; SEM = standard error of the mean; DM = dry matter; LW = liveweight.

**Table 3 animals-12-00822-t003:** Nutrient intake of growing rumen cannulated Nellore steers grazing *Uruchloa brizantha* cv Marandu pastures and fed corn or citrus pulp-based supplements during the wet-dry transition period ^1^.

	Treatment			
Component	Corn	Citrus Pulp	SEM	*p*-Value
Dry matter intake (kg/day)	6.44	6.45	0.09	0.99
Organic matter intake (kg/day)	5.74	5.83	0.08	0.80
Neutral detergent fibre intake (kg/day)	3.53	3.62	0.08	0.68
Crude protein intake (kg/day)	1.05	1.03	0.02	0.56
Total digestible nutrients ^2^ intake (kg/day)	3.89	3.79	0.08	0.62

^1^ Mean values from collections occurring across all experimental periods; ^2^ Total digestible nutrients estimated according to National Research Council [8]; SEM = standard error of the mean.

**Table 4 animals-12-00822-t004:** Nitrogen balance, efficiency of microbial synthesis and blood urea nitrogen of rumen-cannulated Nellore steers grazing *Uruchloa brizantha* cv Marandu pastures and fed corn or citrus pulp-based supplements during the wet-dry transition period ^1^.

	Treatment			
Component	Corn	Citrus Pulp	SEM	*p*-Value
Nitrogen intake (g/day)	167.71	163.86	20.75	0.56
Faecal nitrogen (g/day)	69.69	71.18	2.93	0.70
Urinary nitrogen (g/day)	59.34	55.39	1.12	0.50
Absorbed nitrogen (g/day)	98.03	92.69	0.01	0.45
Retained nitrogen (g/day)	38.68	37.29	0.25	0.99
Retained N: N intake	22.54	22.97	0.34	0.85
Absorbed N: N intake	58.12	56.85	0.13	0.52
Microbial protein synthesis (g/day)	284.44	315.00	0.88	0.41
Microbial nitrogen (g/day)	45.51	50.40	0.48	0.41
EMS ^2^ kg/TDN	77.19	81.18	0.55	0.40
Blood urea nitrogen (mg/dL)	17.80	14.75	0.05	<0.001

^1^ Mean values from collections occurring across all experimental periods, ^2^ Efficiency of microbial synthesis per kilogram of total digestible nutrient intake (g/day); SEM = Standard error of the mean.

**Table 5 animals-12-00822-t005:** Nutrient intake and in vivo digestibility of rumen cannulated Nellore steers grazing *Uruchloa brizantha* cv Marandu pastures and fed corn or citrus pulp-based supplements during the wet-dry transition period ^1^.

	Treatment			
Component	Corn	Citrus Pulp	SEM	*p*-Value
Intake (kg/day)				
Dry matter	7.27	7.68	0.03	0.35
Forage	5.43	5.85	0.07	0.25
Supplement	1.84	1.84	0.01	0.69
Organic matter	6.46	6.94	0.05	0.25
Crude protein	1.20	1.18	0.03	0.98
Ether extract	0.13	0.11	0.04	0.17
Neutral detergent fibre	4.02	4.30	0.04	0.28
Non-fibrous carbohydrate	1.66	1.91	0.03	0.03
Total digestible nutrients ^2^	4.34	4.47	0.03	0.61
Intake (% Liveweight)				
Dry matter	2.07	2.17	0.03	0.35
Forage	1.55	1.65	0.03	0.24
Supplement	0.52	0.52	0.05	0.72
Digestibility in vivo (%)				
Dry matter	54.55	56.18	1.55	0.26
Organic matter	58.09	59.36	2.64	0.40
Ether extract	36.25	31.61	0.03	0.29
Crude protein	62.08	60.72	0.02	0.56
Non-fibrous carbohydrate	87.95	81.42	0.01	0.02
Neutral detergent fibre	49.28	53.74	2.45	0.01

^1^ Mean values from collections occurring across all experimental periods, ^2^ Total digestible nutrients estimated according to National Research Council [8]; SEM = standard error of the mean.

**Table 6 animals-12-00822-t006:** Liveweight performance of Nellore bulls fed corn or citrus pulp-based supplements and beef produced per area in *Uruchloa brizantha* cv Marandu pastures during the wet-dry transition period.

	Treatment			
Component	Corn	Citrus Pulp	SEM	*p*-Value
Initial liveweight (kg)	307.00	305.86	1.23	0.50
Average daily gain (kg/day) ^1^	0.753	0.769	0.03	0.76
Final liveweight (kg)	370.20	368.42	2.05	0.64
Gain per hectare (kg LW gain per ha/day) ^1^	2.71	2.79	0.18	0.85

^1^ Mean values across all experimental periods, SEM = Standard error mean.

## Data Availability

All data supporting reported results can be shared upon request directly to our corresponding author. Complete contact details at https://unesp.br/portaldocentes/docentes/92922?lang=en, accessed on 27 January 2022.

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
