# Peer review of "Citrus Pulp Replacing Corn in the Supplement Decreased Fibre Digestibility with No Impacts on Performance of Cattle Grazing Marandu Palisade Grass in the Wet-Dry Transition Period"

_animals, 2022, doi:10.3390/ani12070822_

Round 1

Reviewer 1 Report

Manuscript animals-1595169, entitled “Performance and metabolism of supplemented cattle grazing marandu palisade grass in the wet-dry transition period”

Recommendation:       The above paper is not suitable for publication in its present form.

General comment

The article provides useful information about the differences in performance and metabolism of corn vs citrus pulp supplemented cattle grazing marandu palisade grass in the wet-dry transition period. Although, the experiment was in general appropriately designed and implemented, there are some points that should be corrected or clarified.

Major comments

  • The title does not correspond to your experimental design. Please rephrase
  • Please provide the three periods of exp.1, since the number of days was not equal.
  • Dry matter intake was evaluated using 36 animals. Not 42? Why?
  • What is the meaning of “**” in the column of SEM in Table 2 and 6?
  • Please explain in detail how the presented data occurred. For example, in Table 4, these data refer to the whole 9-days period? One sample per day or per 9 days?
  • L412-413, 418-420: Where are these data shown?

Minor points

L11: “Deterioration of forage quality as the dry season approaches has detrimental effects…”

L12: “implemented” instead of “put in place”

L18: “applied” instead of “put in place”

L29: “neither” instead of “it did not”

L31: “neither” instead of “no”

L31: Can citrus pulp supplementation have an effect on pastures productivity?

L37: A clear distinction based on?

L40: “period” instead of “moment”

L41: “…and for a feedlot finishing period”? Please rephrase.

L42: “…direct effects on quality…”

L71-72: “…with a total of 188, 28 and 16 kg of nitrogen, phosphorus, and potassium per hectare, respectively, resulting…”

L73: “…dry matter (DM), 15.45% crude protein…”

L75: “in the present study” instead of “here”

L78: Please delete “supplemental”

L92: How many 1.3 ha and 1.0 ha? 6 and 6, respectively?

L93: How was the nutrient intake evaluated?

L102: 4 periods? Please check the 5th general comment.

L105-109: Could you please explain the usefulness of this treatment?

L112: “The collection of forage was performed at the start…”

L118: “transferred” instead of “taken”

L127-128: “…were carried out according to AOAC (1996). Crude…” Please check reference style and use number in [  ].

L131, 138, 143, 161: “…described by…”

L139: “In addition to the aforementioned analyses, the…”

L140: “performed” instead of “done”

L155: “carried out” instead of “done”

L192: How were “Urinary N” measured?

L208: “achieved: instead of “done”

L210: “deprivation” instead of “curfew”

L226-230: How many animals in total? 50?

L276: 4 is too small sample size for DM intake.

L347: “…had an effect…”

L349: “…has been indicated…”

L359: “observed” instead of “found”

L373-374: Please rephrase

L387: “…supplemented with citrus…”

L400: Please delete “both”

L406-407: “…were below it, and as a result ADG observed in exp 1 conducted concomitantly was also lower. This can…”

L408: “…forage allowance; the high…”

L412: “voluntary” instead of “volunteer”

L414, 449: “due to” instead of “because of”

Author Response

Reviewer 1

General comment

The article provides useful information about the differences in performance and metabolism of corn vs citrus pulp supplemented cattle grazing marandu palisade grass in the wet-dry transition period. Although, the experiment was in general appropriately designed and implemented, there are some points that should be corrected or clarified.

 ***Response: Thank you very much for your input! We have addressed all the points risen to improve the overall quality of the final material.

Major comments 

  • The title does not correspond to your experimental design. Please rephrase

***Response: Title has been modified to reflect the results. It now reads: “Citrus pulp replacing corn in the supplement decreased fibre digestibility with no impacts on performance of cattle grazing marandu palisade grass in the wet-dry transition period.

  • Please provide the three periods of exp.1, since the number of days was not equal.

***Response: In exp 1, the three periods were between LW measurements on 15/05/2011 to 10/06/2011 (Period 1), 11/06/2011 to 10/07/2011 (Period 2) and 11/07/2011 to 07/08/2011 (Period 3). This information is now included in lines 120-123.

  • Dry matter intake was evaluated using 36 animals. Not 42? Why?

***Response: That is correct, we’ve utilised the 36 animals. This number was chosen to allow 6 repetitions per block, what was deemed sufficient by the statisticians in our department. It is important to highlight that there were four animals in paddocks of 1.3 ha and three animals in the 1.0 ha paddocks (42 in total). The approach used for the DMI analysis resulted in the same number of repetitions per block and 5 degrees of freedom. To clarify this, the following has been added to lines 232-233: “Dry matter intake was evaluated in the second period of exp 1 using 36 animals, i.e. six repetitions per block, and three markers…”

  • What is the meaning of “**” in the column of SEM in Table 2 and 6?

***Response: The use of the symbol was linked to SEM values lower than 0.001. This was described in the footnote of the tables, however, to avoid confusion the symbols were removed and the revised version of this manuscript now clearly states < 0.001 in the body of the tables.

  • Please explain in detail how the presented data occurred. For example, in Table 4, these data refer to the whole 9-days period? One sample per day or per 9 days?

***Response: The data shown in the Tables is the average between experimental periods. The number of samples collected varied according to variables being studied, what is explained in the M&M section. For example blood was collected once per period on day 12, whilst rumen fluid was collected on day 13, at 0, 3, 6, and 18 h after supplementation. To clarify what has been done, the following has been added to lines 161-162: “with collections occurring across all experimental periods”. In addition, this information is also listed as footnotes in all tables.

  • L412-413, 418-420: Where are these data shown?

***Response: L412-413: We were referring to a lower indigestible NDF of the citrus pulp supplement presented in Table 1 and lower in vivo NDF digestibility in Table 5. To clarify this, the new revised paragraph in lines 431-434 now reads: “Despite the lower iNDF in the citrus pulp-based supplement and lower in vivo NDF digestibility in cannulated steers fed corn, there were no effects on DMI between supplemental treatments”.

 ***Response: 418-420: This sentence is now removed from the revised version of the manuscript.

 Minor points

 L11: “Deterioration of forage quality as the dry season approaches has detrimental effects…”

***Response: Modification done as suggested.

L12: “implemented” instead of “put in place”

***Response: Modification done as suggested.

L18: “applied” instead of “put in place”

***Response: Modification done as suggested.

L29: “neither” instead of “it did not”

***Response: Modification done as suggested.

L31: “neither” instead of “no”

***Response: Modification done as suggested.

L31: Can citrus pulp supplementation have an effect on pastures productivity?

***Response: The use supplements may affect forage intake (e.g. substitution effect), indirectly affecting pastures productivity. In this study there were no differences between supplement types. To clarify this, the sentence has been rewritten. Lines 36-37 now say: “…. on animal performance nor indirect effects on pastures productivity.”

L37: A clear distinction based on?

***Response: A clear distinction on the quantity and quality of forage produced and available to animals. To clarify this, the sentence has been rewritten and lines 43-44 now say: “…. because there is a clear distinction in quantity as well as quality of forage produced and available in these contrasting seasons.”

L40: “period” instead of “moment”

***Response: Modification done as suggested.

L41: “…and for a feedlot finishing period”? Please rephrase.

***Response: The sentence has been rewritten. Now lines 47-48 say: “…. start adapting animals to supplemental feedstuffs and to a feedlot finishing phase.”

L42: “…direct effects on quality…”

***Response: The sentence was missing the word “have”. It now says on lines 49-50: “have direct effects to quality and quantity of the final diet”.

L71-72: “…with a total of 188, 28 and 16 kg of nitrogen, phosphorus, and potassium per hectare, respectively, resulting…”

***Response: Modification done as suggested.

L73: “…dry matter (DM), 15.45% crude protein…”

***Response: Modification done as suggested.

L75: “in the present study” instead of “here”

***Response: Modification done as suggested.

L78: Please delete “supplemental”

***Response: Modification done as suggested.

L92: How many 1.3 ha and 1.0 ha? 6 and 6, respectively?

***Response: That is correct. The sentence has been rewritten. Lines 99-100 now read: “… 12 paddocks (four animals into six paddocks of 1.3 ha and three animals in six 1.0 ha paddocks)…”.

L93: How was the nutrient intake evaluated?

***Response: The sentence has been rewritten to say “TDN intake”. Lines 156-157 say: “Total digestible nutrient contents of forage and supplement samples were estimated according to National Research Council [13].”

L102: 4 periods? Please check the 5th general comment.

***Response: Three periods. This information is now included in lines 120-123.The sentence has been rewritten to indicate the 4 was referring to number of steers used as replicates in the LS design.

L105-109: Could you please explain the usefulness of this treatment?

***Response: In the performance study (Exp 1), occurring parallel to the metabolism trial (Exp 2), animals were fed as a group, with each paddock representing the experimental unit (six in total per treatment). In Exp 2, each of the eight cannulated animals (four per treatment) represented the EU, therefore, we chose to feed them separate to avoid issues of one animal consuming residual supplement left in the bunk.

L112: “The collection of forage was performed at the start…”

***Response: Modification done as suggested.

L118: “transferred” instead of “taken”

***Response: Modification done as suggested.

L127-128: “…were carried out according to AOAC (1996). Crude…” Please check reference style and use number in [  ].

***Response: Modification done as suggested.

L131, 138, 143, 161: “…described by…”

***Response: Modifications done as suggested.

L139: “In addition to the aforementioned analyses, the…”

***Response: Modification done as suggested.

L140: “performed” instead of “done”

***Response: Modification done as suggested.

L155: “carried out” instead of “done”

***Response: Modification done as suggested.

L192: How were “Urinary N” measured?

***Response: Urinary N was estimated using spot urine samples. To clarify this, the paragraph has been rewritten and lines 206-208 now say: “This method was adapted in the present study using spot urine sampling to estimate urinary excretion of nitrogenous compounds and the following equation to estimate N balance:”

L208: “achieved: instead of “done”

***Response: Modification done as suggested.

L210: “deprivation” instead of “curfew”

***Response: Modification done as suggested.

L226-230: How many animals in total? 50?

***Response: That’s correct. The sentence has been rewritten. Line 245 now says: “ADGs of animals in a particular treatment (i.e. 25 per treatment) and…”

L276: 4 is too small sample size for DM intake.

***Response: DM intake was calculated using the intact animals, with 18 replicates per treatment. Information has been added to clarify this. Lines 232-233 now read: “Dry matter intake was evaluated in the second period of exp 1 using 36 animals, i.e. six repetitions per block…”

L347: “…had an effect…”

***Response: Modification done as suggested.

L349: “…has been indicated…”

***Response: Modification done as suggested.

L359: “observed” instead of “found”

***Response: Modification done as suggested.

L373-374: Please rephrase

***Response: The sentence has been rewritten. Now lines 392-393 say: “that observed that microorganisms that use pectin use rumen ammonia N faster than microorganisms that use cellulose or hemicellulose as substrates…”

L387: “…supplemented with citrus…”

***Response: Modification done as suggested.

L400: Please delete “both”

***Response: Modification done as suggested.

L406-407: “…were below it, and as a result ADG observed in exp 1 conducted concomitantly was also lower. This can…”

***Response: Modification done as suggested.

L408: “…forage allowance; the high…”

***Response: Modification done as suggested.

L412: “voluntary” instead of “volunteer”

***Response: Modification done as suggested.

L414, 449: “due to” instead of “because of”

***Response: Modification done as suggested.

Reviewer 2 Report

The study is overall well presented and I have only few comments and/or minor changes to suggest:

Line128. Correct citation and add  reference (in the reference list) for AOAC, 1996 

Lines 125-126: why forages for determination of chemical composition were hand-plucked?

Line 157: how did you know which faeces was from which steer?

Line 172: how long after collection samples were centrifuged ?

Line 176:add reference for factor used

Line 228: correct " according the evaluation period" with “according to the evaluation period”

Line 331: correct “A” with “As” planed

Author Response

The study is overall well presented and I have only few comments and/or minor changes to suggest:

***Response: Thank you! Your input is appreciated.

Line128. Correct citation and add  reference (in the reference list) for AOAC, 1996

***Response: Modification done as suggested.

Lines 125-126: why forages for determination of chemical composition were hand-plucked?

***Response: The idea of using hand-plucked samples is to simulate what an animal would be consuming. Alternatively some authors utilise what is referred to as “grazing horizon”. In this second approach, the forage between a specific height chosen as the grazing management is collected. To clarify this, the following sentence has been added to lines 175-177: “The use of hand-plucked samples was adopted to simulate what an animal would be consuming as indicated by Barbero et al. [8].”

Line 157: how did you know which faeces was from which steer?

***Response: Through visual observations. To clarify this, the sentence has been rewritten. Now lines 167-168 say: “The faeces collections occurred when the animals were observed defecating. The collections…”

Line 172: how long after collection samples were centrifuged ?

***Response: The following has been added to line 188 to clarify this: “and were placed in ice for approximately 20 minutes until processing.”

Line 176: add reference for factor used

***Response: The factor used corresponds to the percentage of N present in the urea molecule. To clarify this, the following has been added at the end of the sentence in lines 192-193: “plasma by 0.4667, which corresponds to the percentage of N present in the urea molecule.”

Line 228: correct " according the evaluation period" with “according to the evaluation period”

***Response: Modification done as suggested.

Line 331: correct “A” with “As” planed

***Response: Modification done as suggested. 

Reviewer 3 Report

Dear authors,

Thank you for this piece of work. I read it with some difficulty which however doesn't shadow any doubts on the quality of the experiments. I honestly reckon  that on the basis of all data you report probably there is room for more than one paper.

This being said, the manuscript is uneasy to read. Sentences are definitely too long with too many different concepts. Experiments should be better splitted as to results which are difficult to contextualize (I would even suggest to write two different papers, maybe part A and part B).

Overall, the discussion is more than once (I mean, in several tracts of the text) shifting from the core of your results, sometime it is too long, sometime it is inappropriate because not sticking to results. Finally, very few is explained from animal side, which is almost totally missing though you narrowed the speech to a very well defined physiological moment (so, not only pasture and feed).I would suggest to compare and maybe discuss on findings and approach reported by Cappai et al., 2019 Res Vet Sci doi: 10.1016/j.rvsc.2018.12.016.

Author Response

Reviewer 3

Dear authors,

Thank you for this piece of work. I read it with some difficulty which however doesn't shadow any doubts on the quality of the experiments. I honestly reckon  that on the basis of all data you report probably there is room for more than one paper.

This being said, the manuscript is uneasy to read. Sentences are definitely too long with too many different concepts. Experiments should be better splitted as to results which are difficult to contextualize (I would even suggest to write two different papers, maybe part A and part B).

Overall, the discussion is more than once (I mean, in several tracts of the text) shifting from the core of your results, sometime it is too long, sometime it is inappropriate because not sticking to results. Finally, very few is explained from animal side, which is almost totally missing though you narrowed the speech to a very well defined physiological moment (so, not only pasture and feed).I would suggest to compare and maybe discuss on findings and approach reported by Cappai et al., 2019 Res Vet Sci doi: 10.1016/j.rvsc.2018.12.016.

***Response: We thank the reviewer for the input. We appreciate the fact that the two experiments could be published separately. However, there was always the intention of running them concomitantly to help us understand the changes in metabolic parameters in parallel to what was happening to animal performance in the field. As seen in our results, there were minor differences in metabolism, and these did not reflect on changes on animal performance. We believe this is a very important outcome, indicating that citrus pulp, a by-product of the citrus industry, can be used in substitution of corn when formulating supplements for cattle grazing in the wet-dry transition period.

Following your recommendation, we have added a sentence to our discussion session. Now lines 529-531 read: “As indicated by Cappai et al. [45] circulating parameters of clinical importance may indicate homeostasis perturbations. In the current study, animal performance was evaluated in parallel to key metabolic parameters ...”

We do appreciate that as you say, the paper could be uneasy to read at times. We have added the input from all reviewers and believe the new revised material is clearer and hopefully a more pleasant read.

Round 2

Reviewer 1 Report

Authors made all the necessary amendments. I suggest the acceptance of their manuscript. However, one last correction should be made. Please check that in Table 2 and 6, the application of "*", "**" or "***" referring to P<0.05, 0.01 or 0.001, respectively, is frequent for P-value. It is not used for SEM. Please provide the numbers as they are.

Author Response

Reviewer 1

Authors made all the necessary amendments. I suggest the acceptance of their manuscript. However, one last correction should be made. Please check that in Table 2 and 6, the application of "*", "**" or "***" referring to P<0.05, 0.01 or 0.001, respectively, is frequent for P-value. It is not used for SEM. Please provide the numbers as they are.

 ***Response: Thank you very much! Your input was greatly appreciated. It allowed us to make important improvements on the final version of this manuscript. As requested, Tables 2 and 6, lines 289-292 and 353-356, respectively, have been checked. Corrections were made where appropriate and the values of SEM have been provided in full.

Reviewer 3 Report

Dear Authors 

Thank you for welcoming almost all requests of changes. The paper improved from the original version, however I suggest still some minor revision and the proofread by a fluent speaker.

Meanwhile M&M were in fact splitted into a more comprehensible fashion, at least to allow the understanding of objectives of exp. 1 and 2, likewise results, discussion is still an overall speech. As said previously, the animal side is totally missing. The citation  of the paper from Cappai et al was important to stress on the fact that not N balance during transition is important, but also body fluid distribution linked to urea is important in the animals which first calve and then start lactation. Neglecting appropriate fluid shift from body compartments would compromise the homeostasis. I would have expected some comments of the quality of forages kin the dry period and the mineral components.

Some minor request of change:

L. 430: 'non-nutritional', please specify.

L. 464: 'The diet nutrient digestibility' please change j to 'Nutrient digestibility'. The text contains similar expressions throughout, so please, I warmly recommend to make the paper proofread.

L. 502-522: Please, make an effort to apply the concept of body fluid distribution.

Thanks.

Author Response

Dear Authors

Thank you for welcoming almost all requests of changes. The paper improved from the original version, however I suggest still some minor revision and the proofread by a fluent speaker.

Meanwhile M&M were in fact splitted into a more comprehensible fashion, at least to allow the understanding of objectives of exp. 1 and 2, likewise results, discussion is still an overall speech. As said previously, the animal side is totally missing. The citation of the paper from Cappai et al was important to stress on the fact that not N balance during transition is important, but also body fluid distribution linked to urea is important in the animals which first calve and then start lactation. Neglecting appropriate fluid shift from body compartments would compromise the homeostasis. I would have expected some comments of the quality of forages kin the dry period and the mineral components.

 ***Response: Thank you! We appreciate your input. We have made changes to accommodate the instructions from all reviewers, which included the addition of the reference by Cappai as you suggested. We believe that it indeed highlighted the importance of specific metabolites circulating the body of a ruminant animal and how these can affect homeostasis.

We would like to highlight that this manuscript has been proofread by a fluent speaker. The corresponding author, Dr Costa, has lived, studied, and worked in English speaking countries for over a decade. Dr Costa went to high school in Ohio and worked for Kansas State University in USA. In addition, Dr Costa has done a PhD in animal nutrition at the University of Queensland in Australia, where he currently resides. Dr. Costa is a senior researcher at Central Queensland University and uses English on a daily basis as part of his professional career. Despite the latter comment, we are happy to modify the current English used in the manuscript from British to American style if required (or preferred).

It is important to highlight that the discussions about forage quality were not emphasised in this manuscript because the basal diet was the same across all treatments. However, to comply with the reviewer’s suggestion, more information has been added to lines 363-366 that now read: “animals were offered a basal diet (marandu palisade grass) of similar characteristics. The crude protein and the TDN content of the forages were not different between treatments and the pastures had equal green leaf allowances.”

Following the reviewer’s suggestions, we attempted to give more depth into the discussion about the animal performance in the current study. Now lines 423-438 read: “In the current study, animals supplemented at 0.5% LW gained in average 0.76 kg per day, similar to liveweight performance of growing bulls under a rotational grazing system observed by Costa et al. [1], using the same supplement allowance and management based on sward heights of 25 cm at the start and 10 cm post-grazing. Based on the equations of Marcondes et al. [32], grazing Nellore bulls of 350 kg of LW with a DM intake of 2.2 % of LW and consuming about 4.75 kg TDN/day would have ADG of 1.0 kg. The average intake values of DM (2.12 % of LW) and TDN (4.41 kg/day) were below it, and as a result ADG observed in exp 1 was also lower. This can be explained by the lower quality of the pastures in the transition period, which inhibits energy intake despite the high forage allowance; the high stem and dead material proportions most likely affected DM in-take. In the work of Costa et al. [1], differences in forage allowance resulted in approximately 0.1 kg difference in ADG between animals in two treatments fed the same supplement at 0.5% LW. Ground corn fed at 0.0%, 0.3%, 0.6% and 0.9% LW to growing beef cattle under a rotational grazing system of marandu palisade grass resulted in linear increases in ADG, stocking rates and in gain per hectare, or the amount of beef produced per area [3].”

And regarding stocking rates on lines 512-515: “Costa et al. [1] observed a 0.43 AE/ha increase on stocking rates in growing bulls grazing marandu palisade grass pastures during the wet season and fed a citrus-pulp based supplement at 0.5% LW in comparison to non-supplemented animals from the same cohort.”

Some minor request of change:

  1. 430: 'non-nutritional', please specify.

 ***Response: The factors referred in this sentence are characteristics of the sward structure, such as leaf:stem ratio, proportion of dead material, density etc. To further clarify this, lines 445-446 now say: “due to the sward structure and non-nutritional factors, e.g. the density and physical properties of grass stems….”.

  1. 464: 'The diet nutrient digestibility' please change j to 'Nutrient digestibility'. The text contains similar expressions throughout, so please, I warmly recommend to make the paper proofread.

 ***Response: In this sentence we are referring to the digestibility of the diet (i.e. basal forage + supplement), hence why we said “the diet nutrient digestibility”. In order to facilitate the understanding, we have rewritten the paragraph. Now lines 483-484 read: “The nutrient digestibility of the final diet, i.e. basal forage and supplement, did not change between treatments…” In addition, we’ve asked a native speaker to proofread the manuscript. A few minor changes were suggested and are indicated using track changes.

  1. 502-522: Please, make an effort to apply the concept of body fluid distribution.

Thanks.

***Response: We have added more into the topic in an attempt to illustrate how the concept could be applied to nutritional work. Now lines 544-551 read: “As indicated by Cappai et al. [45] circulating parameters of clinical importance may indicate homeostasis perturbations. The latter authors highlighted the importance of body fluid distribution and how it links to nutrient and energy availability to organs and tissues. In their work, the concept was applied to pregnant does and showed differences amongst individuals delivering single or twin kids. In the current work, the supplement type was the factor potentially affecting the metabolic profile of the animals. Therefore, differences in animal performance were evaluated in parallel to key metabolic parameters of relevance to ruminant nutrition.”
